# Attentional adversarial training for few-shot medical image segmentation without annotations

**Buhailiqiemu Awudong**[1,2], **Qi Li**[1,2]*, **Zili Liang**[3], **Lin Tian**[4], **Jingwen Yan**[3,5]

**1** School of Computer Science and Technology, Changchun University of Science and Technology, Changchun, China, **2** Zhongshan Institute of Changchun University of Science and Technology, Zhongshan, China, **3** Department of Electronic Engineering, Shantou University, Shantou, China, **4** Department of Electronics and Engineering, Yili Normal University, Yili, China, **5** Key Laboratory of Intelligent Manufacturing Technology, Ministry of Education, Shantou University, Shantou, China

* liqi@cust.edu.cn

**Data Availability Statement:** In our experiments, we evaluate the generalization ability of the proposed framework PG-Net conducting abdominal organs auto-segmentation by different medical image modalities including an abdominal

## Abstract

Medical image segmentation is a critical application that plays a significant role in clinical research. Despite the fact that many deep neural networks have achieved quite high accuracy in the field of medical image segmentation, there is still a scarcity of annotated labels, making it difficult to train a robust and generalized model. Few-shot learning has the potential to predict new classes that are unseen in training with a few annotations. In this study, a novel few-shot semantic segmentation framework named prototype-based generative adversarial network (PG-Net) is proposed for medical image segmentation without annotations. The proposed PG-Net consists of two subnetworks: the prototype-based segmentation network (P-Net) and the guided evaluation network (G-Net). On one hand, the P-Net as a generator focuses on extracting multi-scale features and local spatial information in order to produce refined predictions with discriminative context between foreground and background. On the other hand, the G-Net as a discriminator, which employs an attention mechanism, further distills the relation knowledge between support and query, and contributes to P-Net producing segmentation masks of query with more similar distributions as support. Hence, the PG-Net can enhance segmentation quality by an adversarial training strategy. Compared to the state-of-the-art (SOTA) few-shot segmentation methods, comparative experiments demonstrate that the proposed PG-Net provides noticeably more robust and prominent generalization ability on different medical image modality datasets, including an abdominal Computed Tomography (CT) dataset and an abdominal Magnetic Resonance Imaging (MRI) dataset.

## Introduction

Automatic segmentation of medical images is of great significance for clinical anatomy and pathological structure research including organ segmentation [1], optic disc segmentation [2], tumor segmentation [3], etc. With the remarkable performance of automatic medical

CT dataset (Abd-CT) and an abdominal MRI dataset (Abd-MRI). Abd-CT: https://doi.org/10.7303/syn3193805 doi: 10.7303/syn3193805. Abd-MRI: https://chaos.grand-challenge.org/Data/.

**Funding:** This work was supported in part by the National Natural Science Foundation of China (grant number: 61672335), the Colleges Innovation Project of Guangdong, China (grant number: 2017KCXTD015), Jilin Provincial Scientific and Technological Development Program (grant number: 20200802004GH), and Intramural funds for academic construction (grant number: 22XKZZ22). The funders had no role in study design, data collection and analysis, decision to publish, or preparation of the manuscript.

**Competing interests:** The authors have declared that no competing interests exist.

segmentation, many practical applications have become available to achieve precise treatment and speedy disease diagnosis [4,5]. Training a robust and efficient enough medical imagery segmentation model always requires considerable supervised data. Nevertheless, the scarcity of high-quality annotated medical images causes huge challenges and burdens to experienced clinical experts for the time-consuming and tedious work of labeling. This problem is further exacerbated by data acquisition in different patients, medical equipment and institutions, which is also more challenging for segmentation models to achieve impressive performance of unseen classes.

Hampered by limited annotated data, many recent studies have focused on the few-shot learning [6–9]. Few-shot learning models first learn the potential representations of the support data with a few annotations and then transfer the knowledge to perform pixel-level predictions on unlabeled query images. In the inference process, the trained few-shot model shows the strong generalization ability to make predictions on new classes which are unseen in training by just distilling the features of limited labeled support images. Although previous works adopted a few-shot learning protocol to process natural images and achieved prominent performance [10], it is challenging for medical images to conduct pixel-wise predictions. Pixel-wise prediction is critical because it provides fine-grained information about each individual pixel in a medical image, allowing for a complete comprehension of spatial relationships within the data. Moreover, training a few-shot medical segmentation model which can generate precise binary masks with only a few labeled data has critical implications for health care and clinical diagnosis.

Most current few-shot segmentation works generally extract both support and query features and then conduct knowledge transfer [7,11–16] or feature matching [17–21] from support to query data. However, many of these models are easily prone to overfitting as transferring a few labeled support images trained by a large deep network to query. To alleviate the need for annotated data, we trained a few-shot learning model by a self-supervised learning (SSL) manner. The prototypical network is usually applied in feature matching methods to distill class-wise discriminative prototypes from support images guiding predictions of query. The prototypes are usually calculated with corresponding mask by masked average pooling operation [21–24]. It is impractical that a single prototype represents sufficient spatial features of the entire object region. Although recent approaches proposed prototype alignment [21], prototype refinement [10,25] and prototype mixture [26] to enhance the representative capacity of prototype, the context information and local spatial features in both the foreground and background are ignored. Furthermore, in medical images, the inhomogeneous distributions between the small foreground and the large background deteriorate and the segmentation regions differ greatly. Another challenge is the binary segmented masks generated from the segmentation network may still deviate from the distributions of support images and the additional ambiguity would be introduced to the network [27]. [28] addresses the distribution matching problem between intra-instance and intra-class similarity distributions in order to obtain the "optimal" parameters. Intra-instance similarity describes the similarity between original samples of a certain anatomical structure and their augmented samples, proving the effectiveness of optimal data augmentation in strengthening few-shot segmentation models. Essentially similar, GANs are also an excellent technique of data augmentation for improving the utilization of data features.

Based on the above concerns, we propose a novel few-shot learning network framework for medical image segmentation, the prototype-based generative adversarial network (PG-Net). The architecture of PG-Net is shown in Fig 1. The proposed network consists of two subnetworks, one is the prototype-based segmentation network (P-Net) to generate dense predictions and another is the mask guided network (G-Net) with an attention-based adversarial learning

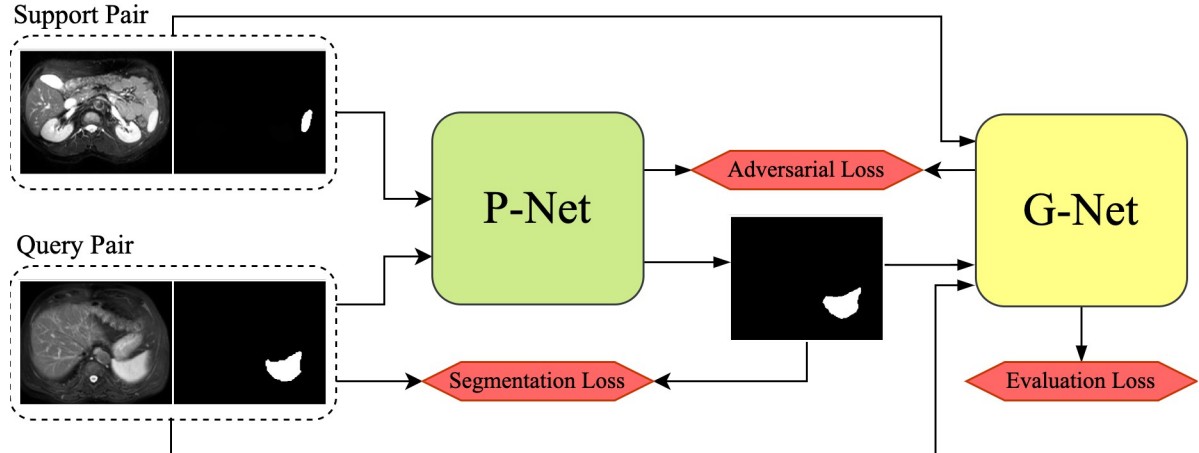

**Fig 1. Proposed PG-Net framework for medical image segmentation based on generative adversarial structure.** It is mainly composed of P- Net and G-Net.

mechanism for guiding P-Net generating less biased segmented mask. The P-Net is mainly composed of three parts including the local prototypes computation, the prediction by prototypes allocation and the assisted segmentation by multi-scale prediction. To obtain sufficient local information both in the foreground and the background, we first compute local spatial prototypes which are representative enough to propaganda between support and query. In medical images, it is crucial to distinguish the segmentation region from rather inhomogeneous context relationships. On one hand, we employ the non-parametric metric learning strategy to enforce the network focus on local similarities between the support and query, which provides specific spatial information to predict on the top of prototypes allocation. On the other hand, the P-Net is encouraged to enhance the discrimination capacity of distinct object region sizes via FPN-like [29] network producing multi-scale prediction maps.

Following the segmentation network P-Net as a generator, G-Net as a discriminator is designed to learn support distributions in order to enable P-Net to generate more precise predictions of query image. The attention-based adversarial learning mechanism effectively extracts the representative features of foreground mask and the support or query images, which contributes to evaluating the quality of query predictions. By constraining the quality of query predictions closed to support distributions, the upstream network P-Net alleviates additional ambiguous and biased features of query. Hence the P-Net is forced to learn to gradually refine the query prediction. Overall, the proposed framework PG-Net combines the core ideas both of the prototype-based few-shot learning and the generative adversarial learning. Moreover, our experiments are conducted in a SSL manner by generating superpixel-based pseudo-labels to alleviate the urgent need for manual labeling.

Our major contributions can be concluded as follows:

1. A novel framework for self-supervised few-shot medical image segmentation is proposed to refine predictions of unlabeled data via an attentional adversarial training strategy.

2. We proposed a prototypical segmentation network that enhances the discrimination of distinct segmentation regions by extracting local spatial information and performing multi-scale prediction maps.

3. The proposed PG-Net is trained in different medical image modalities including an abdomen Computed Tomography (CT) dataset (Abd-CT) and an abdomen Magnetic

Resonance Imaging (MRI) dataset (Abd-MRI) without manual annotations. According to the experiments, the proposed framework outperforms the state-of-the-art self-supervised few-shot learning models for medical image segmentation.

## Related work

### Self-supervised medical image segmentation

Many SSL methods were proposed to alleviate the need for human annotations in various deep learning tasks. Especially in the field of medical image processing, SSL strategies are ingeniously applied in medical image classification, localization and segmentation, etc. SSL focuses on training convolutional neural networks with automatically generated labels. Tomasetti [30] proposed a self-supervised training method specifically designed for ischemic stroke lesion segmentation by utilizing color-coded parametric maps with limited annotated samples. Wang et al. [31] designed a topology clustering network which builds a graph network for transformation-invariant features and conducts modularity maximum clustering on the topology graph to generate pseudo labels for each image. Hang et al. [32] utilized a self-supervised strategy to automatically detect seed regions from spatial domains and build a supervised classifier using temporal information. However, most of these pretext tasks require fine-tuning before solving the specific subsequent tasks.

In the field of superpixel-based segmentation for SSL, many previous works are widely used in the processing of nature images [33–35], but relevant studies on medical images segmentation are rare. Superpixel-based segmentation strategy provides a way of producing pseudo labels with the same objective of follow-up medical image segmentation task. It is desirable to combine the superpixel-based segmentation with medical image processing due to superpixels providing a compact representation of image data by strongly adapting to the spatial structure and grouping similar pixels into clusters. Hansen [36] suggested a few-shot medical image segmentation network based self-supervision task that is motivated by anomaly detection and captures the 3D feature of the data by leveraging supervoxels. In our work, we follow the work of [37] taking advantage of the efficient and unsupervised graph-cut-based algorithm by [38] to produce pseudo labels.

### Few-shot learning

More recent attention has focused on the research of few-shot learning for its significant generalizability to predict unseen classes with limited annotated data. Recent surveys have demonstrated the prominent performance of few-shot learning and show the great potential in various deep learning tasks especially for character recognition [39,40] and image classification [40,41]. However, it is more challenging to apply the methodology of few-shot learning to segmentation tasks for requiring higher-level sematic information to predict in pixel-level. The problem is more prominent in lots of noises and blurry boundaries medical image. Shaban et al. [12] first extended the few-shot classification learning to one-shot segmentation in pixel-level. [13] further focused on the generalized knowledge acquired through classes seen during training to new classes by extracting a latent task representation, from any amount of supervision given. Later on, [17] introduced a method of prototype-based sematic segmentation. Prototypes are illustrated as feature vectors with high-level discriminative information and calculated with pixels in the query image by measuring the similarities to produce prediction map. Following this work, many researchers share a similar spirit of prototype learning. For instance, [42] adopt a masked average pooling strategy for producing guidance features in

support image. [18] proposed a part-aware prototype mechanism which aims to decompose the holistic class representation into a set of part-aware prototypes. [43] proposed a 3D neural network for prototypical cross-institution few-shot multi-class segmentation to solve the challenge of limited data. In these works, the support annotations are used only for masking. Whereas in [21], they introduced a prototype alignment regularization strategy to perform few-shot learning in a reverse direction by the resulted segmentation model.

Nevertheless, almost all the abovementioned works are coped with 2D RGB images and have sufficient annotated support data, which both differ from the medical image segmentation scenarios. And many works on few-shot medical image segmentation require pre-train and fine-tuning for the subsequent mask prediction [44,45]. In contrast, [15] designed a volumetric medical image segmentation strategy that optimally pairs slices of query and support volumes. In a self-supervised setting, a few-shot medical segmentation approach [37] that can generate superpixel-based pseudo labels has attained state-of-the-art performance. Inspired by this work, we conduct further studies to achieve efficient improvement and take this method as one of our comparative experiments.

## Generative adversarial network

A great deal of previous research into semi-supervised learning and unsupervised learning has focused on generative adversarial networks (GANs) for relieving the need for costly and time-consuming human annotations. In addition, GANs has shown its great potential and been widely utilized in various practical application scenarios including style transfer, image synthesis, sequence generation, and semantic segmentation, etc [46].

In medical scenario, [47] generated synthesis images by adding noise vectors and the discriminative network differentiates between the synthetic and real data. [48] extended to designing a cross-modality image synthesis network to learn the mapping between MRI and CT for increasing the amount of training set. Besides these synthetic approaches, some traditional data augmentation operations such as Gaussian blurring, appearance enhancement, spatial transformations are also employed for pre-processing [49,50]. However, GANs methodology based on data augmentation has to face the challenge of deviation and bad fake samples learned by generative networks. Unlike the aforementioned papers, [51] designed an evaluation network to distinguish between segmentation results of unannotated images and annotated ones. That encourages the segmentation network to generate images with more similar features of unlabeled data. [52] also proposed an evaluation network for adversarial learning which adopts a dual-attentive fusion block to distill various levels features from the previous predicted segmentation map.

## Methods

In this section, we first elaborate the problem definition of few-shot medical image segmentation. Then the proposed PG-Net is explained and the subnetworks are also clarified in detail. Next we introduce the design of the loss function.

### Problem definition

In few-shot medical image segmentation task, the key target is to train a model that has great generalization ability to segment new classes totally unseen in the training process. In other words, given a few labeled samples of unseen classes in the inference stage, the model can perform mask prediction. Specifically, sets of semantic classes $C_{tr}$ (e.g., $C_{tr}$ = {liver,spleen}) and $C_{te}$ (e.g., $C_{te}$ = {rightkidney,leftkidney} are given, the training set $D_{tr}$ and the testing set $D_{te}$ can be constructed by these sets that do not intersect respectively ($C_{tr} \cap C_{te} = \emptyset$). In the few-shot

learning, an episodic training strategy is used widely. Every episode is consisted of a set of support images and a set of query images with its corresponding binary masks in the form of data pair $(S,Q)$. Namely, $D_{tr} = \left\{ (I_s, M_s^c)^i; (I_q, M_q^c) \right\}^N$ is composed of N episodes, where $N = 1,2,3,\ldots,n$ denotes the number of episodes and $i = 1,2,3,\ldots,K$ is the index of image-mask pairs from support. The superscript $c = 1,2,3,\ldots,C \in C_{tr}$ is the class index of training set, and the subscripts $s$ and $q$ are defined as support and query data respectively. In particular, every episode is defined as a N-way K-shot problem adopted from [12]. And works of great majority in medical segmentation, 1-way 1-shot task is always carried out [15,37], same as our work. In the inference process, the model performs segmentations by learned from several samples of support in $C_{te}$ without any finetuning and re-training.

## Proposed network

Our proposed model is composed of two subnetworks, the P-Net and the G-Net. The aim of segmentation network is to obtain a model that has adequate generalization capabilities to perform predictions on unseen classes. Considering the problem of lacking sufficient labeled data in medical imagery, the proposed P-Net adopts a prototypes extraction method of few-shot learning. Following the P-Net, G-Net is designed to evaluate the performance of mask prediction from the P-Net. In addition, the G-Net contributes to guiding the P-Net to generate more accurate binary masks. More details of the proposed network are elaborated as follows.

**Segmentation network.** The proposed P-Net is inspired by ALP-Net [37] and ASG-Net [19]. ALP-Net fully exploits spatial information of medical images by average pooling with a specific pooling window to obtain local prototypes, which is different from the previous works [21,25] that use mask-level average pooling ignoring the location relationships of intra-classes. ASG-Net performs multi-level mask predictions with refined features for guiding better mask predictions. Based on the method mentioned above, we proposed the prototype-based segmentation network named P-Net. The framework of P-Net is shown in Fig 2. Specifically, P-Net not only considers carefully to produces spatial local prototypes but also merges multi-level features to perform auxiliary mask predictions guiding the segmentation network to output refined mask.

Given support feature $F_s \in \mathbb{R}^{(Ch \times H \times W)}$, support mask $M_s^c \in \mathbb{R}^{(H \times W)}$, where $(H,W)$ is the spatial size and $Ch$ is the channel depth. $P = \left\{ \mathcal{P}_{loc/glo}^{c,k} \right\}$ is an ensemble comprised the local and global prototypes, where $k$ is the prototype index. First, P-Net takes support masks $M_s^c$ and extracted features $F_s$ of support images for calculating both class-level and local-level

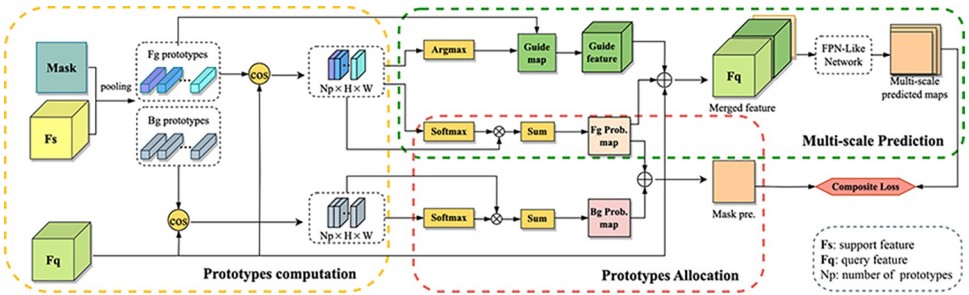

**Fig 2. Workflow of P-Net. P-Net contains three subnetworks**: (1) A prototypes computation subnetwork which adopts a non-parametric metric learning method calculates the cosine similarities for producing foreground and background prototypes. (2) A prototypes allocation subnetwork focuses on local prototypes allocation and fusion in class-level generating foreground and background probability maps. (3) A multi-scale prediction subnetwork performs multi-scale predictions by feeding the merged feature into FPN-Like Network.

prototypes of foreground classes $c$ and background $c_0$. Each local prototype of $\mathcal{P}_{loc}^{c,k}$ can be computed by average pooling with a local pooling window size $(\alpha H, \alpha W)$ defined as $S_w$, where $\alpha$ denotes the window scale, as follows:

$$\mathcal{P}_{loc}^{c,k} = \frac{1}{\alpha^2 HW} \sum_{\alpha H} \sum_{\alpha W} F_s(a,b), \text{ where } (a,b) \in S_w \tag{1}$$

Note that the background prototypes and foreground prototypes have the same average pooling window size. Same as [37], we also apply a masked average pooling to compute a class-level prototype when none of prototypes smaller than pooling window size:

$$\mathcal{P}_{glo}^{c,k} = \frac{\sum\limits_{(x,y)\in(H,W)} F_s(x,y) M_s^c(x,y)}{\sum\limits_{(x,y)\in(H,W)} M_s^c(x,y)} \tag{2}$$

Hence, the local prototypes and the global prototypes involve abundant spatial representation information of intra-classes.

Based on the computed prototypes, we employ a non-parametric metric learning method. We calculate the cosine distances $D^k$ to measure the similarities between the query feature maps $F_q \in \mathbb{R}^{(Ch \times H \times W)}$ and the computed prototypes $P$ at spatial location $(x,y)$ in pixel-level:

$$D^k(x,y) = \cos(F_q(x,y), P) = \alpha \frac{F_q(x,y)\mathcal{P}_{loc/glo}^{c,k}}{\|F_q(x,y)\|_2 \|\mathcal{P}_{loc/glo}^{c,k}\|_2} \tag{3}$$

where $\alpha$ is a scaling factor set to 20 same as [21].

To obtain the final dense predictions, there is a two-branch framework including the prediction of prototypes allocation and the multi-scale prediction.

In the prediction of prototypes allocation, the softmax function is applied after (3) to acquire probability maps $PM \in \mathbb{R}^{(H \times W)}$, that means the local prototypes $\mathcal{P}_{loc}^{c,k}$ are self-adaptively fused to specific semantic classes as a whole or the background class $c_0$, then we get:

$$PM(x,y) = \sum_k D^k(x,y) \text{Softmax}[D^k(x,y)] \tag{4}$$

Hence, we have $PM^c$ indicates the foreground probability map, and $PM^{c_0}$ for the background probability map. In the multi-scale prediction branch, instead of integrating all the local prototypes as a whole, we calculate the most similar prototype in $D^k(x,y)$ at pixel-level as follow:

$$G(x,y) = \text{Argmax} D^k(x,y) \tag{5}$$

Then the guide map is expanded to guide feature maps by placing corresponding prototype at each pixel location. Subsequently, guide feature maps $F_G \in \mathbb{R}^{(Ch \times H \times W)}$, probability map $PM^c$ and query feature maps $F_q$ are concatenate to the merged feature:

$$F_m = F_G \oplus PM^c \oplus F_q \tag{6}$$

where $\oplus$ indicates the concatenation operation along channel dimension.

Finally, the FPN-like network [19,29] takes $F_m$ as input for producing multi-scale binary mask predictions, which contributes to interaction and propaganda of different scale information. The FPN-like network yields a segmentation result in each scale with the top-down structure as shown in Fig 3. Note that the prediction of multi-scale prediction subnetwork is

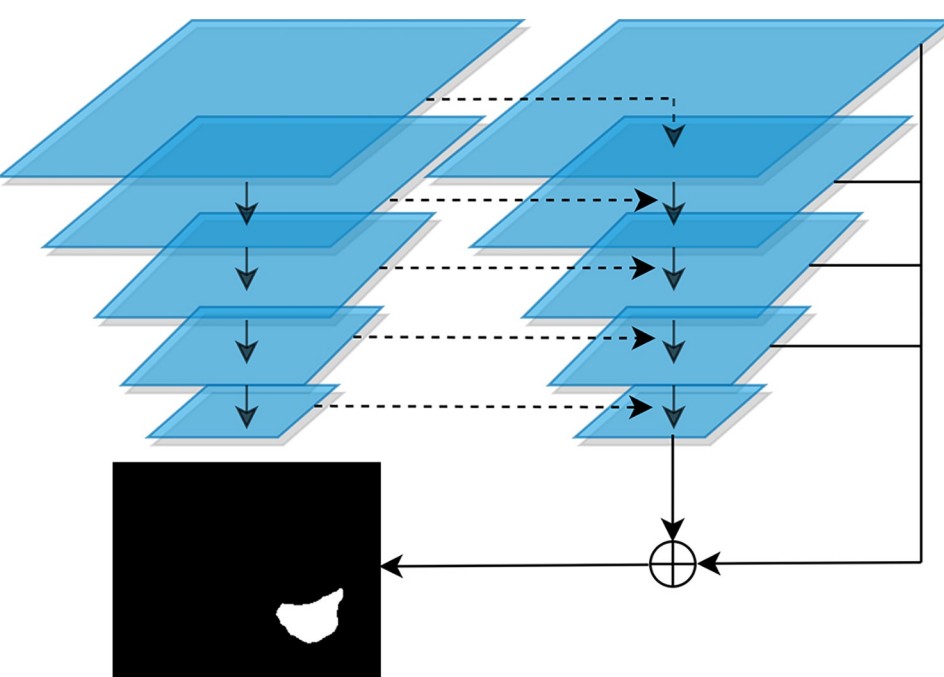

**Fig 3. Structure of FPN-Like network.**

considered as the auxiliary segmentation for averting the introduction of redundant semantic information. And we design a composite loss function to fuse the predictions of the two-branch network in subsection 3. Therefore, the final augmented prediction is given by:

$$m(x, y) = \underset{c}{\mathrm{Argmax}} \; PM(x, y) \qquad (7)$$

## Evaluation network

G-Net aims to distinguish the distributions of predicted masks from the upstream P-Net close to or deviated from masks of support. In other words, the G-Net provides guiding information that allows P-Net to generate dense predictions leveraging masks of query data with more similar distributions. Both the support data and the query data in the proposed self-supervised approach are devoid of human annotations, and all labels are generated by the superpixel-based segmentation method in SSL. As a result, the G-Net learns the relatively accurate mapping relationship between the support image and its superpixel mask for providing guiding information in order to improve query segmentation.

Referring to Fig 4(a), the inputs of G-Net consist of three parts including the original medical images, the corresponding mask and the inverse mask of it. Original images are first fed into a Dense block [53] to extract features. Then the attention block calculates the attentive maps of the mask and the inverse mask with corresponding extracted image feature respectively. Subsequently, the attentive maps are concatenated to dual-attentive maps that are input to successive Dense blocks for generating high-level feature maps. Finally, the last three layers can be regarded as a classifier for transferring high-level features to a binary score. In particular, score 1 indicates good quality whereas 0 for poor quality.

It is crucial for medical image segmentation to enhance robust performance by recovering more boundary features with both foreground and background masks. Similar with [52], we adopt the idea of a dual-attentive fusion strategy that fully utilizes the limited supervised

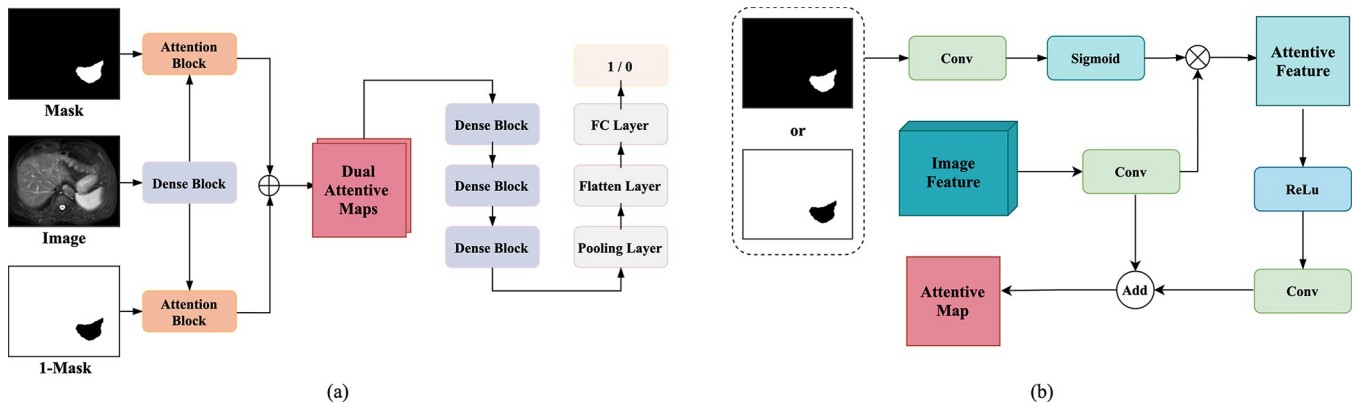

**Fig 4.** (a) Structure of G-Net. (b) Workflow of attention block in G-Net.

information including the segmented region and the background. As illustrated in Fig 4(B), the segmented mask or that of an inverse mask is initially fed into a convolution layer followed by a sigmoid layer to acquire informative features, which are then multiplied with extracted image features in element-wise along with channels. The attentive feature map is further refined by the convolution layer and then added with query feature map in pixel-level for obtaining the final fused attentive feature map. In general, the dual-attentive fusion strategy not only effectively improves the discernibility of the G-Net, but also contributes to converging to an ideal statue for the P-Net.

## Loss function

In the proposed network, the training process is conducted with an end-to-end learning strategy. During training, an episode $(S_i, Q_i)$ is input in each iteration to form a 1-way 1-shot segmentation scenario. The $Seg(\cdot)$ and the $Eva(\cdot)$ indicates the P-Net and the G-Net respectively.

For each iteration of P-Net, we design a composite loss function composed of the main segmentation loss $\mathcal{L}_{seg}$ and the adversarial $\mathcal{L}_{adv}$, which is presented as:

$$\mathcal{L}_P = \mathcal{L}_{seg} + \lambda_0 \mathcal{L}_{adv} \tag{8}$$

where $\lambda_0$ indicates the weight of the adversarial learning, and when $\lambda_0$ is set to 0.02, the performance achieves better. We extensively discussed the selection of hyperparameters and determining the best hyperparameter configuration based on test set performance using grid search methodology.

The main segmentation loss $\mathcal{L}_{seg}$ measures the pixel-level deviation between the predicted mask $\hat{M}_q$ obtained from $Seg(\cdot)$ and the pseudo-label $M_q$ by minimizing three subitem loss functions as following:

$$\mathcal{L}_{seg} = \ell_{pro} + \ell_{align} + \lambda_1 \ell_{fpn} \tag{9}$$

where $\ell_{pro}$ is the loss function of the cosine-similarity-based non-parametric metric learning algorithm and is computed by cross-entropy loss, hence we have:

$$\ell_{pro}(I_q, M_q) = \ell_{ce}(Seg(I_q), M_q) = -\frac{1}{HW} \sum_{(x,y) \in (H,W)} M_q \log \hat{M}_q \tag{10}$$

The prototypical alignment regularization (PAR) methodology proposed in [21,37] is also employed to our experiment for further assisting prediction with by exploiting more available support features. Specifically, the predicted mask $\hat{M}_q$ and the corresponding pseudo-label $M_q$ consist of the new support data $\tilde{S}_i$, while the origin support image will be segmented. Hence the new pair $(\tilde{S}_i, Q_i)$ is input to the network to compute $\ell_{dign}$ written as:

$$\ell_{align}(I_s, M_s) = \ell_{ce}(Seg(I_s), M_s) = -\frac{1}{HW} \sum_{(x,y)\in(H,W)} M_s \log \hat{M}_s \tag{11}$$

where $\hat{M}_q$ is the prediction of $M_q$ taking $I_s$ as query.

As for the FPN-like network in the P-Net, each scale performs a segmentation map $y_{fpn}^l$ for computing the loss $\ell_{fpn}$, where the superscript $l$ is the index of scale. The loss function is referred as:

$$\ell_{fpn}(I_q, M_q) = \ell_{ce}(Seg(I_q), M_q) = -\frac{1}{HW} \sum_{(x,y)\in(H,W)} \sum_l M_q \log y_{fpn}^l \tag{12}$$

Note that $\lambda_1$ is a constant regulating the strength of multi-scale prediction loss and when setting to 0.3, the network performed well. The adversarial loss $\ell_{adv}$ is the binary cross-entropy loss function, determining to further improve prediction by reducing the distribution deviation between the segmented map and the label, and is defined as:

$$\mathcal{L}_{adv} = \ell_{bce}(Eva(I_q, M_q), 1) = -\frac{1}{HW} \sum_{(x,y)\in(H,W)} M_q \log \hat{M}_q + (1 - M_q) \log(1 - \hat{M}_q) \tag{13}$$

With the aim of generating segmentation maps with closer distributions as pseudo-label of support, we design the loss function of G-Net represented as:

$$\mathcal{L}_G = \mathcal{L}_S + \lambda_2 \mathcal{L}_Q \tag{14}$$

where $\lambda_2$ is the loss coefficients set to 0.5. Both $\mathcal{L}_S$ and $\mathcal{L}_Q$ loss functions are calculated by binary cross-entropy $\ell_{bce}$ same as that in (13), defined as following:

$$\mathcal{L}_S(I_s, M_s) = \ell_{bce}(Eva(I_s, M_s), 1) \tag{15}$$

$$\mathcal{L}_Q(I_q, M_q) = \ell_{bce}(Eva(I_q, Seg(I_q)), 0) \tag{16}$$

## Experiments

On the basis of an abdominal CT dataset and an abdominal MRI dataset, we compare the proposed PG-Net with recent state-of-the-art methods and conduct some ablation experiments to illustrate the effectiveness of PG-Net. The experimental details as well as the evaluation metric are shown.

### Dataset

In our experiments, we evaluate the generalization ability of the proposed framework PG-Net conducting abdominal organs auto-segmentation by different medical image modalities including an abdominal CT dataset (Abd-CT) and an abdominal MRI dataset (Abd-MRI).

1. Abd-CT: Abd-CT is a clinical abdomen CT dataset from the MICCAI 2015 Multi-Atlas Abdomen Labeling challenge [54]. It contains 30 3D abdominal CT scans from patients with various pathologies and variations in intensity distributions between scans. Although

this dataset is intrinsically complex, it offers extensive information outside of their regions-of-interest, which helps SSL by supplying superpixel sources.

2. Abd-MRI: Abd-MRI is an abdomen MRI dataset from ISBI 2019 Combined Healthy Abdominal Organ Segmentation Challenge [55]. It includes 20 3D T2-SPIR MRI scans, which broadens the imaging modalities under consideration and boosts the rigor of our study.

## Implementation details

In the preprocessing, a superpixel-based pseudo labels generation method is also applied in our experiments to address the scarcity of manual annotations following the same protocol as previous work setting [37]. For fair comparison, our experiments are conducted by 5-fold cross validation under 1-way 1-shot setting. Within each fold, we choose two unseen semantic classes (e.g. right kidney and left kidney) for testing and the other semantic classes (e.g. liver and spleen) for training. In addition, the slices which contain the unseen semantic classes of test set are eliminated for training a generalized few-shot learning model. In the few-shot learning scenarios, the model is normally trained with 2D images. To fit into few-shot learning framework, all 3D volumetric images are segmented into 2D slices with a cropped size of $256 \times 256$ in our experiments. Following the volumetric segmentation strategy used in [15,37], both support and query volume are divided into 9 chunks. Each query slice and the center slice of corresponding support chunk consist of support-query pairs. The initial weights of our proposed framework are pretrained in MS-COCO with an effective backbone of ResNet101 [56] as a feature extractor in order to obtain high-level extracted feature maps of both query and support images. The PG-Net is trained in an end-to-end manner for 80 epochs, which utilizes the stochastic gradient descent as optimizer with initial learning rate 0.001 decayed by 0.98 per epoch. All experiments are carried out on computer servers with one GPU card (NVIDIA GeForce GTX TITAN XP) and the network is implemented with PyTorch.

## Evaluation metric

The evaluation metric Dice similarity coefficient (DSC) is widely used in medical image segmentation scenario. The DSC quantifies the overlapping pixel regions between the segmentation results and the true labels, with values ranging from 0 to 1. "1" indicates that the segmentation result completely overlaps with the real label. We also use the DSC to measure the similarity between prediction map $X$ and ground truth $Y$. The DSC is written as:

$$DSC = \frac{2|X \cap Y|}{|X| + |Y|} \times 100\% \tag{17}$$

## Comparison with state-of-the-art methods

We compare the proposed network PG-Net with previous works on few-shot segmentation scenario respectively. In order to show the performance and effectiveness of the proposed PG-Net, the following state-of-the-art methods are selected as the comparison, and the comparison results are shown in Table 1.

SE-Net [15] is the first applying few-shot learning methodology to volumetric medical image segmentation with an ingenious volumetric segmentation strategy that optimally pairs the slices of query and support volumes. PANet [21] is a typical prototype-based network

**Table 1. Comparative experimental results in DSC (%) between the PG-Net and other state-of-the-art methods.**

| Datasets | Method | Manual Annotation | Right Kidney | Left Kidney | Spleen | Liver | Mean |
|---|---|---|---|---|---|---|---|
| Abd-CT | SE-Net [15] | ✓ | 14.34 | 32.83 | 0.23 | 0.27 | 11.91 |
| | PANet [21] | ✓ | 17.37 | 32.34 | 25.59 | 38.42 | 29.42 |
| | RP-Net [22] | ✓ | 70.00 | 70.48 | 69.85 | 79.62 | 72.48 |
| | SSL-PANet [37] | × | 34.69 | 37.58 | 27.73 | 47.37 | 35.11 |
| | SSL-ALPNet [37] | × | 54.82 | 63.34 | 60.25 | 73.65 | 63.02 |
| | AD-Net [27] | × | 56.98 | 63.84 | 61.84 | 73.95 | 64.15 |
| | Self-ref+ [28] | × | 62.12 | 68.06 | **67.64** | 73.70 | 67.88 |
| | PG-Net (Ours) | × | **64.89** | **69.28** | 67.28 | **79.06** | **70.13** |
| | Fully supervised [57] | ✓ | 92.0 | 95.3 | 96.8 | 97.4 | 95.4 |
| Abd-MRI | SE-Net [15] | ✓ | 61.32 | 62.11 | 51.8 | 27.43 | 50.66 |
| | PANet [21] | ✓ | 38.64 | 53.45 | 50.90 | 42.26 | 46.33 |
| | RP-Net [22] | ✓ | 85.78 | 81.40 | 76.35 | 73.51 | 79.26 |
| | SSL-PANet [37] | × | 47.95 | 47.71 | 58.73 | 64.99 | 54.85 |
| | SSL-ALPNet [37] | × | 78.39 | 73.63 | 67.02 | 73.05 | 73.02 |
| | AD-Net [27] | × | 76.02 | 71.89 | 65.84 | 76.03 | 72.70 |
| | Self-ref+ [28] | × | 80.73 | **77.85** | **69.79** | 73.96 | 75.58 |
| | PG-Net (Ours) | × | **81.78** | 77.08 | 66.97 | **79.58** | **76.35** |
| | Fully supervised [58] | ✓ | - | - | - | - | 94.6 |

Note: The best result without manual annotation in each column is in boldface.

which learns class-specific prototype representations and introduces a prototype alignment regularization. RP-Net [22] is also a prototypical network which enhances the context relationship feature by context relation encoders and achieves a SOTA recently.

However, the above-mentioned methods take manual annotations as input to train a generative few-shot model, which is insufficiently comparable to our proposed framework trained with unlabeled data.

SSL-ALPNet [37] is the first work that explores SSL for few-shot medical image segmentation. SSL-PANet [37] is a modified PANet which also employs the superpixel-based SSL strategy. Superpixel-based SSL shows the great potential of encouraging few-shot models to learn generalizable features and was applied a number of studies for image preprocessing, which is added to our proposed model.AD-Net [27] is an anomaly detection-inspired approach to few-shot medical image segmentation, which only consider foreground prototypes to avoid the challenges associated with explicitly modeling the large and varied background class. It uses a single foreground prototype to compute anomaly ratings for all query pixels. Self-ref + [28] is an optimal few-shot medical image segmentation model without annotation by aligning the intra-instance and intra-class similarity distribution.

Compared with the methods that without using manual annotations, PG-Net achieves a DSC of 70.13% for Abd-CT dataset, outperforming Self-ref + by 2.25%. And for Abd-MRI dataset, PG-Net's DSC is 76.35%, surpassing Self-ref +'s score of 75.58%. In contrast, compared with previous proposed models trained with labeled data, PG-Net gains a higher DSC than PANet by a large margin about 40% and 30% on Abd-CT and Abd-MRI respectively, but is slightly inferior to RP-Net by 2.35% and 2.91% on Abd-CT and Abd-MRI respectively.

From these comparative experiments analysis, our proposed PG-Net shows the prominent generalization ability and can achieve the SOTA on diverse medical image modalities of both Abd-CT and Abd-MRI. Although current few-shot medical segmentation methods are less effective than fully supervised methods, few-shot learning shows the great potential on generalization ability of unseen classes with limited labeled data or no annotations, such as SSL-ALPNet and PG-Net. PG-Net outperforms SSL-ALPNet for the distinctive segmentation region by effective local and multi-scale representation extraction. As shown in Fig 5, the proposed

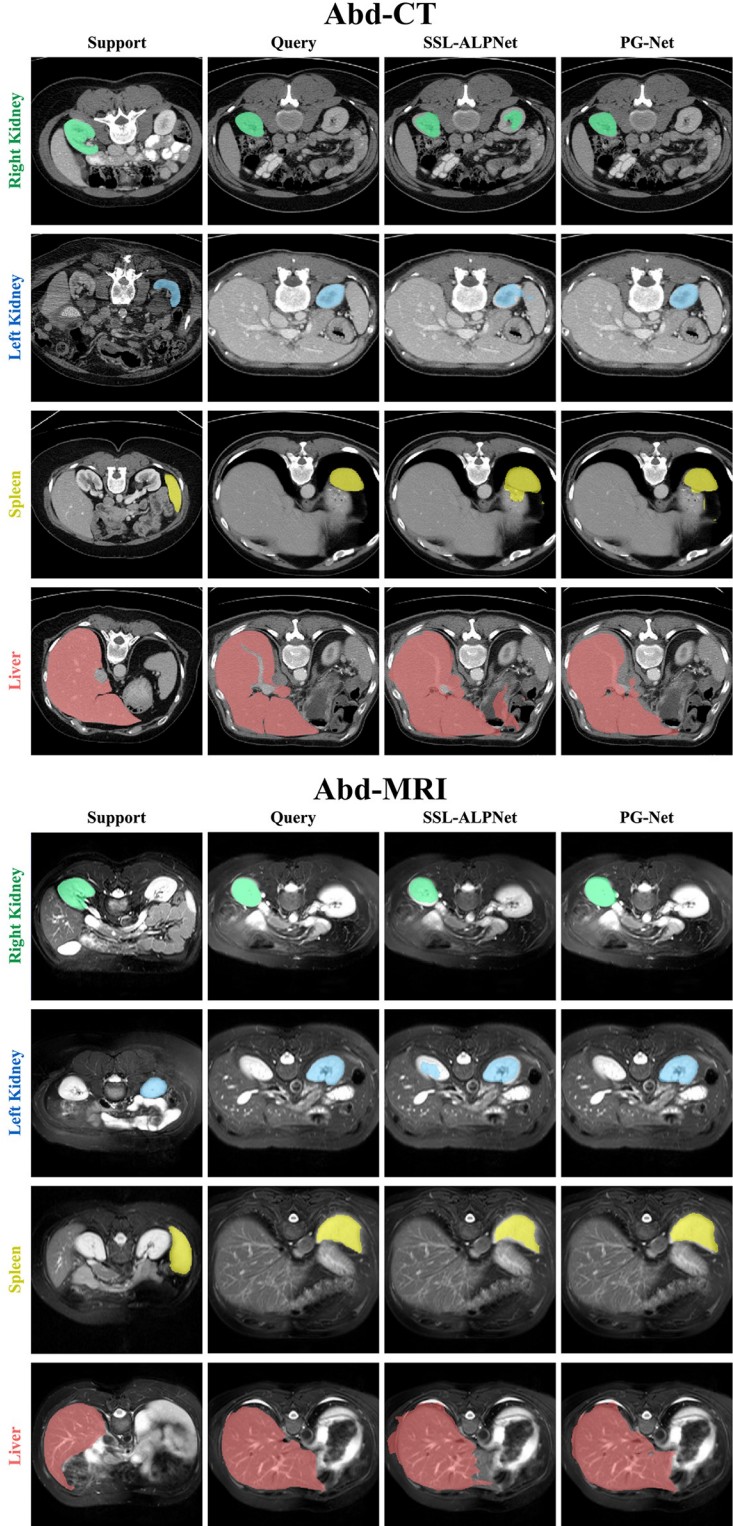

**Fig 5. Qualitative results of our model in 1-way 1-shot segmentation on Abd-CT and Abd-MRI.**

**Table 2. Experimental results of different components of proposed PG-Net on Abd-CT dataset.**

| Added components | Right Kidney | Left Kidney | Spleen | Liver | Mean |
|---|---|---|---|---|---|
| Multi-scale Prediction | 43.31 | 49.31 | 50.06 | 69.24 | 52.98 |
| Prototypes Allocation | 56.94 | 65.00 | 63.43 | 77.14 | 65.63 |
| P-Net | 58.56 | 65.34 | 65.28 | 77.47 | 66.66 |
| P-Net + G-Net (Single Path Attention) | 64.23 | 68.74 | 66.23 | 78.20 | 69.35 |
| P-Net + G-Net (Dual-Attentive-Fusion) | **64.89** | **69.28** | **67.28** | **79.06** | **70.13** |

Note: The best result in each column is in boldface.

network generates satisfying segmentation on abdominal organs compared with the SOTA method.

## Ablation study

In this section, we conduct ablation study to analyze the effectiveness and contributions of different components of the proposed PG-Net:

1. Multi-scale Prediction: Model trained without G-Net and the segmentation results produced only by multi-scale prediction structure.

2. Prototypes Allocation: Model trained without G-Net and the segmentation results produced only by prototypes allocation architecture.

3. P-Net: Model trained by P-Net which adds multi-scale prediction framework to prototypes allocation framework without G-Net.

4. P-Net + G-Net (Single Path Attention): Model trained by P-Net and G-Net, which only used single-path attention maps of the foreground for training.

5. P-Net + G-Net (Dual-Attentive-Fusion): Model trained by the whole network composed of P-Net and G-Net, which used dual attention paths fusing foreground and background attention maps,

The ablation experiments are carried out on Abd-CT illustrated in Table 2. The DSC scores of different frameworks indicate that these components have prominent improvement to varying degrees. The first two rows display the performance of multi-scale prediction architecture and prototypes allocation subnetwork respectively. Although multi-scale prediction's performance is unsatisfactory, combining it with the prototypes allocation framework results in a higher DSC score (displayed in third row). Note that the model trained only by prototypes allocation outperforms the SSL-ALPNet illustrated in Table 1 by 2.61%. The last two rows indicate that further adding the G-Net to the P-Net, the segmentation accuracy can further achieve improvement (The use of a Single Path Attention based generative adversarial network model can improve the DSC score by 2.69% compared to P-Net, while the use of a complete Dual Path Attention PG-Net model can increase 3.47%).

## Conclusion

In this work, we proposed a novel few-shot medical segmentation framework PG-Net using prototypical segmentation network based on generative adversarial network architecture. PG-Net is composed of two subnetworks. The P-Net learns the local spatial representations and performs multi-scale prediction maps, which enhances the discrimination of segmentation region. And the G-Net can evaluate the quality of prediction and learn the correct

distributions of query for encouraging the P-Net to produce refined mask with homogeneous context. In addition, our proposed model is trained without any manual annotations and shows the great potential of generalization ability to new classes, which is of great significance of medical image processing. Furthermore, the architectures of multi-scale prediction and generative adversarial networks can be easily extended to other few-shot segmentation networks. There are several limitations to this study. Firstly only 2D medical images can be used with PG-Net. Subsequent studies will investigate its use in 3D medical image segmentation. Second, because different organ tissues differ structurally, applying the model to more modalities and diverse organ segmentation datasets for further improving the model's generalization and adaptability.

## Acknowledgments

The authors would like to thank all those who provided raw data.

## Author Contributions

**Conceptualization:** Buhailiqiemu Awudong, Zili Liang, Jingwen Yan.

**Data curation:** Zili Liang.

**Formal analysis:** Buhailiqiemu Awudong, Lin Tian.

**Funding acquisition:** Lin Tian, Jingwen Yan.

**Investigation:** Buhailiqiemu Awudong, Jingwen Yan.

**Methodology:** Buhailiqiemu Awudong, Zili Liang.

**Project administration:** Qi Li, Jingwen Yan.

**Resources:** Qi Li, Jingwen Yan.

**Software:** Zili Liang, Lin Tian.

**Supervision:** Qi Li, Jingwen Yan.

**Validation:** Buhailiqiemu Awudong.

**Visualization:** Buhailiqiemu Awudong, Zili Liang.

**Writing – original draft:** Buhailiqiemu Awudong.

**Writing – review & editing:** Buhailiqiemu Awudong, Qi Li, Zili Liang, Lin Tian, Jingwen Yan.

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
