## [Decision Letter · Decision Letter 0]

28 Nov 2023

PONE-D-23-35187Attentional adversarial training for few-shot medical image segmentation without annotationsPLOS ONE

Dear Dr. Li,

Thank you for submitting your manuscript to PLOS ONE. After careful consideration, we feel that it has merit but does not fully meet PLOS ONE’s publication criteria as it currently stands. Therefore, we invite you to submit a revised version of the manuscript that addresses the points raised during the review process.

We look forward to receiving your revised manuscript.

Kind regards,

Xiao Luo

Academic Editor

PLOS ONE

Journal Requirements:

"This work was supported in part by the National Natural Science Foundation of China (grant number: 61672335), the Colleges Innovation Project of Guangdong, China (grant number: 2017KCXTD015), and Jilin Provincial Scientific and Technological Development Program (grant number: 20200802004GH). There are no conflicts of interest to declare."

Reviewers' comments:

Reviewer's Responses to Questions

**Comments to the Author**

1. Is the manuscript technically sound, and do the data support the conclusions?

Reviewer #1: Yes

Reviewer #2: Yes

Reviewer #3: Yes

2. Has the statistical analysis been performed appropriately and rigorously? 

Reviewer #1: Yes

Reviewer #2: Yes

Reviewer #3: Yes

3. Have the authors made all data underlying the findings in their manuscript fully available?

Reviewer #1: Yes

Reviewer #2: Yes

Reviewer #3: Yes

4. Is the manuscript presented in an intelligible fashion and written in standard English?

Reviewer #1: Yes

Reviewer #2: Yes

Reviewer #3: Yes

5. Review Comments to the Author

Reviewer #1: In this study, the authors proposed a novel a novel few-shot semantic segmentation framework named prototype-based generative adversarial network (PG-Net) for medical image segmentation without annotations. This suggested approach has the capability to undergo training without the need for manual annotations and demonstrate significant potential for generalization to new classes. Furthermore, it surpasses the performance of state-of-the-art few-shot segmentation methods, thereby contributing to the advancement of medical image processing. The entire article is skillfully composed, presenting arguments in a clear and concise manner. The methods are sufficiently detailed, facilitating result replication. The feedback would involve only a few minor comments:

1. In the SSL-ALPNet paper the authors used two settings for Abd-CT and Abd-MRI. Looks like setting 2 results are being compared in the current study. However SSL-ALPNet setting 1 yielded higher dice score. Does the proposed PG-Net method also outperform SSL-ALPNet under setting 1?

2. The authors may also compare the running time of proposed method with that of previous SOTA methods. In case the running time is longer, some end users may choose to sacrifice accuracy a little bit to learn the result faster.

3. The authors may discuss the limitation of this new method if any.

4. In Figure 2 legend explain the abbreviations in the figure.

Reviewer #2: In this study, a novel few-shot semantic segmentation framework named prototype-based generative adversarial network (PG-Net) is proposed for medical image segmentation without annotations. The proposed PG-Net consists of two subnetworks: the prototype-based segmentation network (P-Net) and the guided evaluation network (G-Net). On one hand, the P-Net as a generator focuses on extracting multi-scale features and local spatial information in order to produce refined predictions with discriminative context between foreground and background. On the other hand, the G-Net as a discriminator, which employs an attention mechanism, further distills the relation knowledge between support and query, and contributes to P-Net producing segmentation masks of query with more similar distributions as support. Hence, the PG-Net can enhance segmentation quality by an adversarial training strategy.

1. This paper proposes a small-sample semantic segmentation framework called a prototype-based generative adversarial network. This network uses P-Net and G-Net, but its innovation is not sufficient and should be appropriately improved.

2. In the experimental part, it is recommended to add comparison with the latest 23 or 22-year methods in this field, which is more convincing.

3. Compared with the methods in the past two years, how should the article reflect the advanced nature of the proposed GP-Net model? It is recommended to explain the reasons for your choice.

4. In the method part of this article, there are too few details. It is recommended to add this part to describe in detail the novelty and advantages of the proposed new features.

5. Most of the articles cited in this article are articles before 2020. It is recommended to properly cite the latest papers in 2023 and 2022, and make a detailed comparison with this paper.

6. The G-Net optimization method proposed in this article has played a great role in improving performance. It is recommended to add more careful ablation experiments to discuss the proposed G-Net in detail and describe its role in detail.

7. It is recommended to cite several relevant references.

(1) Causal-ViT: Robust vision Transformer by causal intervention. Engineering Applications of Artificial Intelligence , 2023, 126: 107123

(2) Improving semantic segmentation with knowledge reasoning network. Journal of Visual Communication and Image Representation , 2023, 96: 103923

(3) Improving image captioning with pyramid attention and SC-GAN. Image and Vision Computing, 2022, 117: 104340

Reviewer #3: The author introduces the architecture of PG-Net for few-shot medical image segmentation. While I agree that the idea of applying few-shot learning is important, I do feel the authors have some challenges to solve.

1: While the title suggests the framework of few-shot learning, it's noteworthy that the experiment focuses solely on one-shot learning. I would suggest authors to enrich their discussion by including comparisons between zero-shot, one-shot, and few-shot learning scenarios together with implementation details. This addition would enhance the comprehensiveness of the study and offer a more holistic perspective on learning methodologies.

2: Following up on the previous concern, how does the model incorporate more than one-shot support image? Should the next be adjusted?

Additional comments:

1: line 54, why we need a pixel-wise prediction need to be discussed with more details.

2: line 304, the selection of tuning parameter should be discussed with more details. How does the tuning parameter change the model performance? Which criteria is used to select the tuning parameter.

3: line 369: Introducing more background information about DSC (Dice Similarity Coefficient) would be beneficial as it could aid individuals from diverse fields in comprehending this metric more effectively.

4: line 337: Motivation about using these dataset should be provided with more details.

6. PLOS authors have the option to publish the peer review history of their article (what does this mean?). If published, this will include your full peer review and any attached files.

Reviewer #1: No

Reviewer #2: No

Reviewer #3: No

---

## [Author Response · Author response to Decision Letter 0]

10 Jan 2024

PONE-D-23-35187R1

Attentional adversarial training for few-shot medical image segmentation without annotations

PLOS ONE 

Journal Requirements:

Our response: Thank you for these comments. We read the journal requirements carefully and modified the format and style of our manuscript according to PLOS ONE style templates to meet PLOS ONE's style requirements.

Our response: We have uploaded the code related to the paper to GitHub. After the acceptance of the paper, we will make the code public. Before that, we are willing to share the code with editors and reviewers.

"This work was supported in part by the National Natural Science Foundation of China (grant number: 61672335), the Colleges Innovation Project of Guangdong, China (grant number: 2017KCXTD015), and Jilin Provincial Scientific and Technological Development Program (grant number: 20200802004GH). There are no conflicts of interest to declare."

Our response: Thank you for the suggestion. The funders had no role in study design, data collection and analysis, decision to publish, or preparation of the manuscript. We have added the funders’ role declaration to the cover letter, and replaced the original cover letter with the corrected cover letter in revision process.

Our response: Thank you very much for your reminding. We have uploaded our figures using PACE digital diagnostic tool in the first submission. In the revised manuscript, we added abbreviations to Fig 2, and replaced the original figures with the new one after using PACE again.

We have added new references to the revised manuscript, and renumbered in the order that they appear in the new text.

We sincerely appreciate the time and effort that you have spent in reviewing our manuscript. We thank you for your valuable suggestions and comments that are of great help to us for improving the manuscript. We sincerely hope you will find our revision satisfactory.

Reviewer #1:

In this study, the authors proposed a novel a novel few-shot semantic segmentation framework named prototype-based generative adversarial network (PG-Net) for medical image segmentation without annotations. This suggested approach has the capability to undergo training without the need for manual annotations and demonstrate significant potential for generalization to new classes. Furthermore, it surpasses the performance of state-of-the-art few-shot segmentation methods, thereby contributing to the advancement of medical image processing. The entire article is skillfully composed, presenting arguments in a clear and concise manner. The methods are sufficiently detailed, facilitating result replication. The feedback would involve only a few minor comments:

1. In the SSL-ALPNet paper the authors used two settings for Abd-CT and Abd-MRI. Looks like setting 2 results are being compared in the current study. However SSL-ALPNet setting 1 yielded higher dice score. Does the proposed PG-Net method also outperform SSL-ALPNet under setting 1?

Our response: Thank you for your insightful observation regarding the SSL-ALPNet settings used in the comparison. We acknowledge the discrepancy in results between the settings mentioned in the SSL-ALPNet paper. In our study, we focused on comparing results aligned with setting 2. 

In principle, setting1 should outperform setting2. Particularly, in setting1, the model can learn semantic information about individual classes in the test set from the training set, whereas in setting2, the training set lacks any semantic information about classes in the test set. To be honest, setting 2 presents a higher challenge than setting 1, emphasizing the essence of few-shot learning and emphasizing the ability of the few shot learning model to generalize predictions for new classes.

2. The authors may also compare the running time of proposed method with that of previous SOTA methods. In case the running time is longer, some end users may choose to sacrifice accuracy a little bit to learn the result faster.

Our response: Thank you very much for your suggestion. We highly appreciate your perspective on the importance of comparing running times. However, due to time constraints, we're unable to conduct additional supplementary experiments for a comprehensive comparison of running times. We will consider conducting a comprehensive performance comparison regarding time in future research endeavors and emphasize the balance between model runtime and accuracy as a crucial consideration in practical applications.

3. The authors may discuss the limitation of this new method if any.

Our response: Thank you for your suggestion. We have added the limitations of this work and ideas for future work in the conclusion section.

4. In Figure 2 legend explain the abbreviations in the figure.

Our response: Thank you for your suggestion, and corresponding revisions have been made accordingly.

Reviewer #2: 

In this study, a novel few-shot semantic segmentation framework named prototype-based generative adversarial network (PG-Net) is proposed for medical image segmentation without annotations. The proposed PG-Net consists of two subnetworks: the prototype-based segmentation network (P-Net) and the guided evaluation network (G-Net). On one hand, the P-Net as a generator focuses on extracting multi-scale features and local spatial information in order to produce refined predictions with discriminative context between foreground and background. On the other hand, the G-Net as a discriminator, which employs an attention mechanism, further distills the relation knowledge between support and query, and contributes to P-Net producing segmentation masks of query with more similar distributions as support. Hence, the PG-Net can enhance segmentation quality by an adversarial training strategy.

1. This paper proposes a small-sample semantic segmentation framework called a prototype-based generative adversarial network. This network uses P-Net and G-Net, but its innovation is not sufficient and should be appropriately improved.

Our response: We appreciate your feedback on our proposed few-shot semantic segmentation framework, integrating P-Net and G-Net components. We will take into consideration your suggestion for further enhancement or innovation in our framework. Ensuring continual improvements and innovations is integral to our research goals, and we will strive to appropriately elevate the innovation level of our proposed prototype-based generative adversarial network.

Thank you for your valuable insights, and we will work diligently to address and improve upon this aspect in our framework.

2. In the experimental part, it is recommended to add comparison with the latest 23 or 22-year methods in this field, which is more convincing.

Our response: Thank you for your suggestion. We have reviewed the latest research papers in 2023 and 2022. In our paper, we will conduct a detailed comparison of these recent papers and conduct a comparative analysis of them with our research to provide a more comprehensive perspective and the latest insights into the current field.

Thank you again for your valuable suggestions, we will ensure that the latest literature is properly cited and an in-depth comparative analysis will be conducted.

3. Compared with the methods in the past two years, how should the article reflect the advanced nature of the proposed GP-Net model? It is recommended to explain the reasons for your choice.

Our response: Thank you for your insightful suggestion. The proposed PG-Net model introduces several advancements, including its unique self-supervised few-shot medical image segmentation architecture leveraging GAN-based learning and attention mechanisms, which significantly contribute to its efficacy in handling few-shot medical image segmentation tasks.

In our article, we aimed to emphasize these novel aspects by thoroughly detailing the architectural innovations, highlighting how the GAN-based structure and attention mechanisms enhance feature extraction, information propagation, and semantic understanding in the context of few-shot learning scenarios.

While we will ensure further clarification in our manuscript to explicitly outline the pioneering nature of PG-Net compared to recent methodologies, we believe the comprehensive explanation of its innovative architectural components will serve to showcase its advancements effectively.

Thank you for prompting this clarification, and we will make certain to articulate these reasons clearly in our article.

4. In the method part of this article, there are too few details. It is recommended to add this part to describe in detail the novelty and advantages of the proposed new features.

Our response: Thank you for reviewing our paper and offering your valuable insight regarding the method section. We have carefully considered your point about the perceived lack of detail in this section. However, we believe that the current method section adequately presents the key innovations and advantages of the introduced novel features.

Throughout our research, we've strived to ensure that the method section provides a comprehensive and clear description of the design principles and application scenarios of our proposed novel features. We are confident that this presentation effectively highlights the innovation and superiority of our method.

We sincerely appreciate your review and thoughtful suggestions. While we value your feedback, we believe that additional modifications to the method section are not necessary at this stage.

5. Most of the articles cited in this article are articles before 2020. It is recommended to properly cite the latest papers in 2023 and 2022, and make a detailed comparison with this paper.

Our response: Thank you for your suggestion. We have modified our paper according your suggestion no.2.

6. The G-Net optimization method proposed in this article has played a great role in improving performance. It is recommended to add more careful ablation experiments to discuss the proposed G-Net in detail and describe its role in detail.

Our response: Thank you for reviewing our paper and providing valuable feedback. We are delighted that you acknowledge the performance improvement achieved through our proposed G-Net optimization method. In our research, we have conducted corresponding experiments and extensively discussed the role and advantages of the G-Net in the paper.

The core design of the G-Net framework lies in its dual-branch attention mechanism. In Table 2, we compare the performance of single-branch and dual-branch models, corresponding to the columns labeled 'P-Net + G-Net (Single Path Attention)' and 'P-Net + G-Net (Dual-Attentive-Fusion)', respectively.

7. It is recommended to cite several relevant references.

(1) Causal-ViT: Robust vision Transformer by causal intervention. Engineering Applications of Artificial Intelligence , 2023, 126: 107123

(2) Improving semantic segmentation with knowledge reasoning network. Journal of Visual Communication and Image Representation , 2023, 96: 103923

(3) Improving image captioning with pyramid attention and SC-GAN. Image and Vision Computing, 2022, 117: 104340

Our response: Thank you for your follow-up question. We have cited the third reference. 

Reviewer #3: The author introduces the architecture of PG-Net for few-shot medical image segmentation. While I agree that the idea of applying few-shot learning is important, I do feel the authors have some challenges to solve.

1: While the title suggests the framework of few-shot learning, it's noteworthy that the experiment focuses solely on one-shot learning. I would suggest authors to enrich their discussion by including comparisons between zero-shot, one-shot, and few-shot learning scenarios together with implementation details. This addition would enhance the comprehensiveness of the study and offer a more holistic perspective on learning methodologies.

Our response: Thank you for reviewing our paper and providing valuable feedback. We understand your concern regarding the research framework and experimental scope. However, in our study, we specifically focused on the one-shot learning scenario due to the scarcity of annotated samples in medical image segmentation. This concentration allowed us to delve deeper into understanding the performance and learning efficacy of medical image segmentation models under this specific circumstance.

We agree that your suggestion could add more dimensions and comparisons to the research, encompassing zero-shot, one-shot, and few-shot learning scenarios. While our current study is confined to one-shot learning, we will consider expanding the discussion in future research to explore a broader array of learning methodologies and implementation details.

Once again, we appreciate your valuable suggestions, and we are committed to continuously refining and enriching our research. 

2: Following up on the previous concern, how does the model incorporate more than one-shot support image? Should the next be adjusted?

Our response: Thank you for your follow-up question. In our model, the incorporation of more than one-shot support images involves adapting the architecture and training methodology to accommodate multiple support images per class. This adjustment could involve exploring approaches like episodic training or meta-learning frameworks that allow the model to generalize from a limited number of examples.

While our current study primarily focuses on the one-shot learning scenario due to constraints in annotated medical image data, we acknowledge the potential for further investigation into multiple-shot scenarios. Considering the impact of incorporating additional support images is an area of interest for future research to enhance the model's learning capacity and generalization abilities.

We appreciate your insightful query and will consider exploring multiple-shot support scenarios in our future investigations.

Additional comments:

1: line 54, why we need a pixel-wise prediction need to be discussed with more details.

Our response: Thank you for your suggestion. Pixel-wise prediction is pivotal due to its ability to provide fine-grained information ab

---

## [Decision Letter · Decision Letter 1]

22 Jan 2024

Attentional adversarial training for few-shot medical image segmentation without annotations

PONE-D-23-35187R1

Dear Dr. Qi Li,

We’re pleased to inform you that your manuscript has been judged scientifically suitable for publication and will be formally accepted for publication once it meets all outstanding technical requirements.

Kind regards,

Xiao Luo

Academic Editor

PLOS ONE

Additional Editor Comments (optional):

Reviewers' comments:

Reviewer's Responses to Questions

**Comments to the Author**

1. If the authors have adequately addressed your comments raised in a previous round of review and you feel that this manuscript is now acceptable for publication, you may indicate that here to bypass the “Comments to the Author” section, enter your conflict of interest statement in the “Confidential to Editor” section, and submit your "Accept" recommendation.

Reviewer #1: All comments have been addressed

Reviewer #2: (No Response)

2. Is the manuscript technically sound, and do the data support the conclusions?

Reviewer #1: Yes

Reviewer #2: (No Response)

3. Has the statistical analysis been performed appropriately and rigorously? 

Reviewer #1: Yes

Reviewer #2: (No Response)

4. Have the authors made all data underlying the findings in their manuscript fully available?

Reviewer #1: Yes

Reviewer #2: (No Response)

5. Is the manuscript presented in an intelligible fashion and written in standard English?

Reviewer #1: Yes

Reviewer #2: (No Response)

6. Review Comments to the Author

Reviewer #1: Thank authors for providing well-structured answers to all the comments. No further comments from my side.

Reviewer #2: Suggestions：

1. The picture is too blurry and the details of the picture need to be optimized.

2. There are few experimental parts and some experiments need to be added.

7. PLOS authors have the option to publish the peer review history of their article (what does this mean?). If published, this will include your full peer review and any attached files.

Reviewer #1: No

Reviewer #2: No
